# UNDERSTANDING CLASSIFIER MISTAKES WITH GENERATIVE MODELS

## ABSTRACT

Although deep neural networks are effective on supervised learning tasks, they have been shown to be brittle. They are prone to overfitting on their training distribution and are easily fooled by small adversarial perturbations. In this paper, we leverage generative models to identify and characterize instances where classifiers fail to generalize. We propose a generative model of the features extracted by a classifier, and show using rigorous hypothesis testing that errors tend to occur when features are assigned low-probability by our model. From this observation, we develop a detection criteria for samples on which a classifier is likely to fail at test time. In particular, we test against three different sources of classification failures: mistakes made on the test set due to poor model generalization, adversarial samples and out-of-distribution samples. Our approach is agnostic to class labels from the training set which makes it applicable to models trained in a semi-supervised way.

## 1 INTRODUCTION

Machine learning algorithms have shown remarkable success in challenging supervised learning tasks such as object classification (He et al., 2016) and speech recognition (Graves et al., 2013). Deep neural networks in particular, have gained traction because of their ability to learn a hierarchical feature representation of their inputs. Neural networks, however, are also known to be brittle. As they require a large number of parameters compared to available data, deep neural networks have a tendency to latch onto spurious statistical dependencies to make their predictions. As a result, they are prone to overfitting and can be fooled by imperceptible adversarial perturbations of their inputs (Szegedy et al., 2013; Kurakin et al., 2016; Madry et al., 2017). Additionally, modern neural networks are poorly calibrated and do not capture model uncertainty well (Gal & Ghahramani, 2016; Kuleshov & Ermon, 2017; Guo et al., 2017). They produce confidence scores that do not represent true probabilities and consequently, often output predictions that are over-confident even when fed with out-of-distribution inputs (Liang et al., 2017). These limitations of neural networks are problematic as they become ubiquitous in applications where safety and reliability is a priority (Levinson et al., 2011; Sun et al., 2015).

Fully probabilistic, generative models could mitigate these issues by improving uncertainty quantification and incorporating prior knowledge (e.g, physical properties (Wu et al., 2015)) into the classification process. While great progress has been made towards designing generative models that can capture high-dimensional objects such as images (Oord et al., 2016a; Salimans et al., 2017), accurate probabilistic modeling of complex, high-dimensional data remains challenging.

Our work aims at providing an understanding of these failure modes under the lens of probabilistic modelling. Instead of directly modeling the inputs, we rely on the ability of neural networks to extract features from high-dimensional data and build a generative model of these low-dimensional features. Because deep neural networks are trained to extract features from which they output classification predictions, we make the assumption that it is possible to detect failure cases from the learned representations.

Given a neural network trained for image classification, we capture the distribution of the learned feature space with a Gaussian Mixture Model (GMM) and use the predicted likelihoods to detect inputs on which the model cannot produce reliable classification results. We show that we are able to not only detect adversarial and out-of-distribution samples, but surprisingly also *identify inputs from*

*the test set on which a model is likely to make a mistake*. We experiment on state-of-the-art neural networks trained on CIFAR-10 and CIFAR-100 (Krizhevsky, 2009) and show, through statistical hypothesis testing, that samples leading to classification failures tend to correspond to features that lie in a low probability region of the feature space.

**Contributions**    Our contributions are as follows:

- We provide a probabilistic explanation to the brittleness of deep neural networks and show that classifiers tend to make mistakes on inputs with low-probability features.

- We demonstrate that a simple modeling by a GMM of the feature space learned by a deep neural network is enough to model the probability space. Other state-of-the-art methods for probabilistic modelling such as VAEs (Kingma & Welling, 2013) and auto-regressive flow models (Papamakarios et al., 2017) fail in that regard.

- We show that generative models trained on the feature space can be used as a single tool to reliably detect different sources of classification failures: test set errors due to poor generalization, adversarial samples and out-of-distribution samples.

## 2    RELATED WORK

An extensive body of work has been focused on understanding the behaviours of deep neural networks when they are faced with inputs on which they fail. We provide a brief overview below:

**Uncertainty quantification**    Uncertainty quantification for neural networks is crucial in order to detect when a model's prediction cannot be trusted. Bayesian approaches (MacKay, 1992; Neal, 2012; Blundell et al., 2015), for example, seek to capture the uncertainty of a network by considering a prior distribution over the model's weights. Training these networks is challenging because the exact posterior is intractable and usually approximated using a variety of methods for posterior inference. Closely related, Deep Ensembles (Lakshminarayanan et al., 2017) and Monte-Carlo Dropout (Gal & Ghahramani, 2016) consider the outputs of multiple models as an alternative way to approximate the distribution. Model calibration (Platt, 1999; Guo et al., 2017) aims at producing confidence score that are representative of the likelihood of correctness. Uncertainty quantification may also be obtained by training the network to provide uncertainty measures. Prior Networks (Malinin & Gales, 2018) model the implicit posterior distribution in the Bayesian approach, DeVries & Taylor (2018); Lee et al. (2017) have the network output an additional confidence output. These methods require a proxy dataset representing the out-of-distribution samples to train their confidence scores.

Our method differs from the above as it seeks to give an uncertainty estimation based on a model trained with the usual cross-entropy loss. It does not require additional modelling assumptions, nor modifications to the model's architecture or training procedure. As such, it relates closely to threshold-based methods. For example, Hendrycks & Gimpel (2016) use the logits outputs as a measure of the network's confidence and can be improved using Temperature Scaling (Guo et al., 2017; Liang et al., 2017), a post-processing method that calibrates the model. Our work derives a confidence score by learning the probability distribution of the feature space and generalizes to adversarial samples (Szegedy et al., 2013), another source of neural networks' brittleness.

**Adversarial samples**    Methods to defend against adversarial examples include explicitly training networks to be more robust to adversarial attacks (Tramèr et al., 2017; Madry et al., 2017; Papernot et al., 2015). Another line of defense comes from the ability to detect adversarial samples at test time. Song et al. (2017) for example, use a generative model trained on the input images to detect and purify adversarial examples at test time using the observation that adversarial samples have lower predicted likelihood under the trained model. Closer to our work, Zheng & Hong (2018) and Lee et al. (2018) train a conditional generative model on the feature space learned by the classifier and derive a confidence score based on the Mahalanobis distance between a test sample and its predicted class representation. Our method makes the GMM class-agnostic, making it applicable to settings where labels are not available at inference time. We further show that the unsupervised GMM improves on the Mahalanobis score on the OOD detection task.

# 3 DETECTING MISTAKES

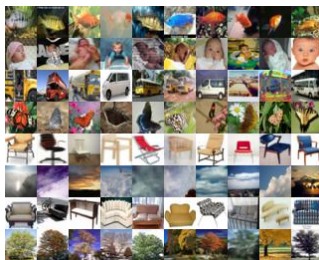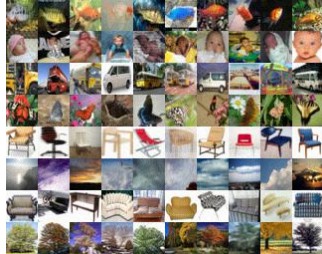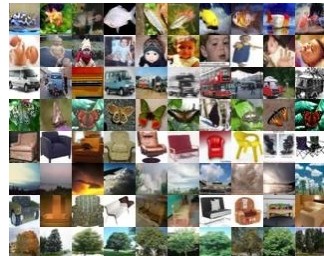

Figure 1: Predicting whether an image will be correctly classified is challenging. Left: Images from the train set. Middle: Adversarial images computed on the same images with FGSM-0.1 are indistinguishable from the clean images, yet, they fool our classifier into making incorrect predictions. Right: Images from the test set. The images look similar to images from the training set, yet, they are incorrectly classified by our DenseNet model. Each row represents a different class.

Detecting samples on which a trained classifier is likely to make a mistake is crucial when considering the range of applications in which these models are deployed. However, predicting in advance whether a sample will fail seems challenging, especially when the sample is drawn from the same distribution as the train set. To illustrate this, we show in Fig. 1, samples from the CIFAR-100 training dataset and compare them to test samples and adversarial examples that our DenseNet model fails to classify properly. In both cases, it is not obvious to the human eye what fundamentally differs between correct and incorrect samples. Our main intuition is that a generative model trained on the feature space could capture these subtle differences.

## 3.1 BACKGROUND

We consider the problem of classification where we have access to a (possibly partially) labeled dataset $\mathcal{D} = \{(\mathbf{X}_i, y_i)\}_{i=1}^N$ where $(\mathbf{X}_i, y_i) \in \mathcal{X} \times \mathcal{Y}$. Samples are assumed to be independently sampled from a distribution $p_{data}(\mathbf{X}, y)$ and we denote the marginal over $\mathbf{X}$ as $p_{data}(\mathbf{X})$. We will denote $f_\theta : \mathcal{X} \to \mathcal{F} = \mathbb{R}^D$ the feature extractor part of our neural network, where $\theta$ represents the parameters of the network and $\mathcal{F}$ is the feature space of dimension $D$. Given an input $\mathbf{X}$, the predictions probabilities on the label space $\mathcal{Y}$ are then typically obtained using multivariate logistic regression on the extracted features.

$$p(y|\mathbf{X}, \theta, \mathbf{W}, \mathbf{b}) = softmax(\mathbf{W}f_\theta(\mathbf{X}) + \mathbf{b}) \tag{1}$$

where $(\mathbf{W}, \mathbf{b})$ represent the weights and bias of the last fully-connected layer of the neural network. The model prediction is the class with the highest predicted probability: $\hat{y}(\mathbf{X}) = \arg\max_{y \in \mathcal{Y}} p(y|\mathbf{X}, \theta, \mathbf{W}, \mathbf{b})$. The parameters $(\theta, \mathbf{W}, \mathbf{b})$ are trained to minimize a cross-entropy loss on the training set and performance is evaluated on the test set.

**Learning the data structure with Generative Models** Understanding the data structure can greatly improve the ability of neural models to generalize. Recently, great progress has been made in designing powerful generative models that can capture high-dimensional complex data such as images. PixelCNN (Salimans et al., 2017; Oord et al., 2016b;a) in particular, is a state-of-the-art deep generative model with tractable likelihood that represents the probability density of an image as a fully factorized product of conditionals over individual pixels of an image.

$$p_{CNN}(\mathbf{X}) = \prod_{i=1}^n p_\phi(\mathbf{X}_i|\mathbf{X}_{1:i-1}) \tag{2}$$

Flow models such as the Masked Autogressive Flow (MAF) (Papamakarios et al., 2017) model provide similar tractability by parameterizing distributions with reversible functions which make that likelihood easily tractable through the change of variable formula. Another widely used class of generative models assumes the existence of unobserved latent variables. Gaussian Mixture Models, for example, assume discrete latents (corresponding to the mixture component). Variational autoencoders (Kingma & Welling, 2013) use continuous latent variables and parameterize the (conditional) distributions using neural networks.

## 3.2 MODELING THE FEATURE SPACE

We identify two main reasons why characterizing examples over which a classifier is likely to make a mistake is difficult. First, modeling the input data distribution $p_{data}(\mathbf{X})$, as done in Song et al. (2017) to detect adversarial examples, is challenging because of the high-dimensional, complex nature of the image space $\mathcal{X}$. This approach also fails at detecting out-of-distribution samples, with state-of-the art models assigning higher likelihoods to samples that completely differ from their train set (Nalisnick et al., 2018). Second, a model of $p_{data}(\mathbf{X})$ doesn't capture any information about the classifier itself.

To overcome these difficulties, we propose to *model the underlying distribution of the learned features* $\mathbf{F} = f_\theta(\mathbf{X})$, where $\mathbf{X} \sim p_{data}(\mathbf{X})$. Extracted features have lower dimension which makes them easier to model and they give access to some information on the classifier. Specifically, we are interested in comparing features $\mathbf{F}_c$ of samples that are correctly classified with features $\mathbf{F}_w$ of samples that are incorrectly classified by a trained neural network. $\mathbf{F}_c$ and $\mathbf{F}_w$ can be described as elements of the following sets:

$$\mathbf{F}_c \in \mathcal{C} = \{f_\theta(\mathbf{X}) | \hat{y}(\mathbf{X}) = y, (\mathbf{X}, y) \in \mathcal{X} \times \mathcal{Y}\} \tag{3}$$

$$\mathbf{F}_w \in \mathcal{W} = \{f_\theta(\mathbf{X}) | \hat{y}(\mathbf{X}) \neq y, (\mathbf{X}, y) \in \mathcal{X} \times \mathcal{Y}\} \tag{4}$$

The distribution of the extracted features is modeled by:

$$p(\mathbf{F}) = \sum_{k=1}^{K} \pi_k \mathcal{N}(\mathbf{F}; \boldsymbol{\mu}_k, \boldsymbol{\Sigma}_k) \tag{5}$$

where $K$ is the number of Gaussians in the mixture, $\pi_k, \boldsymbol{\mu}_k, \boldsymbol{\Sigma}_k$ are the model parameters. We choose $\boldsymbol{\Sigma}_k$ to be diagonal in all our experiments. After training a neural network to convergence, we learn the parameters of the GMM using the EM algorithm. Our training set is built from the features extracted from the training image set by the trained classifier.

## 3.3 DETECTING CLASSIFICATION MISTAKES

We posit that classification mistakes are linked to extracted features that are unusual under the training distribution. By modeling the feature space learned by the classifier, our generative model will be able to detect an input that will lead to a potential classification mistake. We found that a simple generative model is surprisingly good at capturing the distribution of the feature space and can detect when an input will lead to a classification mistake based on its predicted feature log-likelihood.

**Statistical Hypothesis Testing** We consider $p_{\mathcal{C}}(\mathbf{F}_c)$ the distribution of features $\mathbf{F}_c = f_\theta(\mathbf{X})$ where $(\mathbf{X}, y) \sim p_{data}(\mathbf{X}, y)$ and $\hat{y}(\mathbf{X}) = y$, and $p_{\mathcal{W}}(\mathbf{F}_w)$ the distribution of features $\mathbf{F}_w = f_\theta(\mathbf{X})$ where $(\mathbf{X}, y) \sim p_{data}(\mathbf{X}, y)$ and $\hat{y}(\mathbf{X}) \neq y$. These correspond to features extracted on correctly classified vs. incorrectly classified examples. Note that these distributions not only depend on the underlying data distribution but also on the classifier's parameters $(\theta, \mathbf{W}, \mathbf{b})$.

Assuming we have access to samples $\mathbf{F}_{c,1}, \ldots, \mathbf{F}_{c,n} \sim p_{\mathcal{C}}$ and $\mathbf{F}_{w,1}, \ldots \mathbf{F}_{w,m} \sim p_{\mathcal{W}}$ our null hypothesis $H_0$ and alternative hypothesis $H_1$ are:

$$H_0 : p_{\mathcal{C}} = p_{\mathcal{W}} \quad H_1 : p_{\mathcal{C}} \neq p_{\mathcal{W}} \tag{6}$$

We use the Mann-Whitney U-test, which assumes that samples can be ranked. The test statistic is defined by ranking all samples of the two groups together and using the sum of their ranks.

$$U_{\mathcal{C}} = R_{\mathcal{C}} - \frac{n(n+1)}{2} \quad U_{\mathcal{W}} = R_{\mathcal{W}} - \frac{m(m+1)}{2} \tag{7}$$

where $R_{\mathcal{C}}$ and $R_{\mathcal{W}}$ are the sum of ranks of samples $\mathbf{F}_c$ and $\mathbf{F}_w$ respectively. The statistic for the statistical test is $U = \min(U_{\mathcal{C}}, U_{\mathcal{W}})$, which has a distribution that can be approximated by a normal distribution under the null hypothesis. In our approach, samples are ranked based on their predicted probability.

Since our test statistic directly uses the predicted likelihood of a feature, we deduce from it a simple per-sample test to determine if an input is likely to be misclassified. Given a threshold $T$, a test sample $\mathbf{X}$ is rejected as being misclassified if $p(f_\theta(\mathbf{X})) < T$. The value of the threshold is chosen by cross-validation on the validation set to obtain a good trade-off between precision and recall.

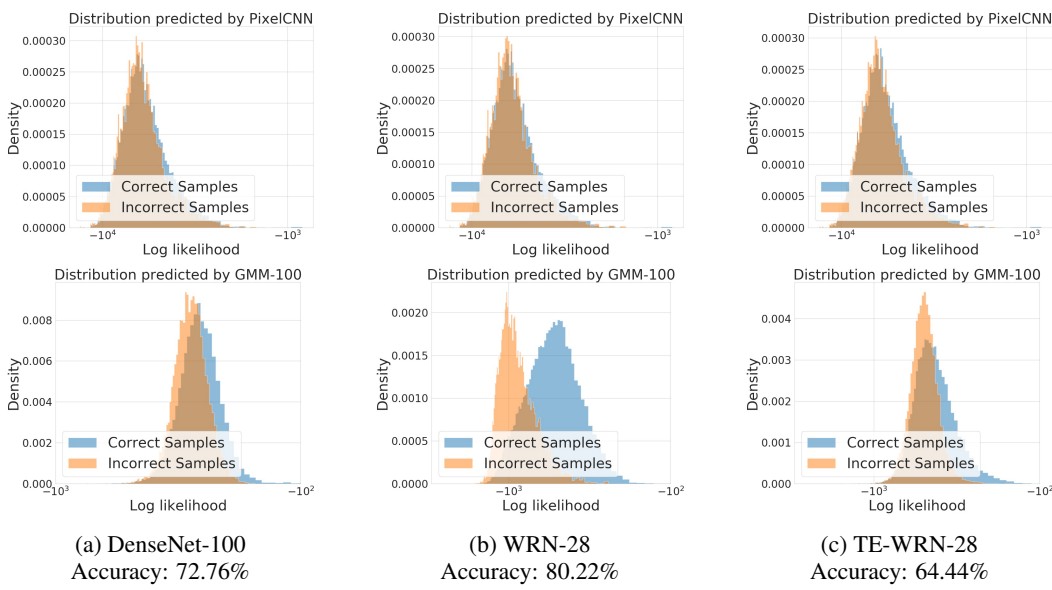

(a) DenseNet-100
Accuracy: 72.76%

(b) WRN-28
Accuracy: 80.22%

(c) TE-WRN-28
Accuracy: 64.44%

Figure 2: Comparing the predicted log likelihood distribution of correct (blue) and incorrect (orange) samples from the test set for models trained on CIFAR-100. Top row: Log likelihoods obtained from training PixelCNN on images from the train set. Bottom row: Log likelihoods obtained from training GMM-100 on features extracted from the train set.

## 4 EXPERIMENTS

We run experiments on the CIFAR-100 dataset, containing $32 \times 32$ color images used for image classification with 100 classes. All reported results give the mean and standard deviation over 5 independent runs. Additional experiments on a model trained on the smaller CIFAR-10 dataset are also available in the appendix. We examine two state-of-the-art deep neural networks, DenseNet-100 (Huang et al., 2016) and Wide ResNet-28 (Zagoruyko & Komodakis, 2016) trained with the usual cross-entropy loss. In the setting where only a small number of labels is available, we train a WRN-28 model with 100 labeled samples per class using Temporal ensembling (Laine & Aila, 2016). This self-ensembling training method takes advantage of the stochasticity provided by the use of dropout and random augmentation techniques (e.g. random flipping and cropping).

**Mistake Detection**     Using statistical testing, we verify that the trained model learns a distribution that differentiates correct and incorrect samples. We sum up the performance of our method by reporting the AUC-ROC and AUC-PR obtained on the test set.

To motivate the use of high-level features, we adapt the detection method used by Song et al. (2017) to the mistake detection problem and compare the performance with our proposed method. We train a PixelCNN on the image dataset and use the predicted likelihood values to detect classification mistakes. We evaluate mistake detection on the test set and first compare the distribution predicted by PixelCNN on the images with the distribution predicted by a GMM-100 model on extracted features (Figure 2). Using the Mann-Whitney U-test, we verify that the distribution learned by GMM-100 differentiates correct and incorrect samples ($p = 1.9e^{-13}$). On the other hand, because PixelCNN is trained without knowledge of the classifier's internal representations, the distributions of correct and incorrect samples predicted under PixelCNN are almost indistinguishable ($p = 8.58e^{-5}$).

Additionally, we experimented with more flexible likelihood models to model the feature space such as the Variational Auto Encoder (Kingma & Welling, 2013) and Masked Autoregressive Flow (Papamakarios et al., 2017). Surprisingly, we found that a simple Gaussian Mixture Model is better at detecting classification mistakes than these more flexible models. Finally, we also compare with other threshold-based methods: using the predicted logits and calibrated scores obtained after Temperature Scaling. Detection performance is summed up in Figure 3 for DenseNet and WideResNet models trained on CIFAR-100. GMM models trained on the features outperform all other generative

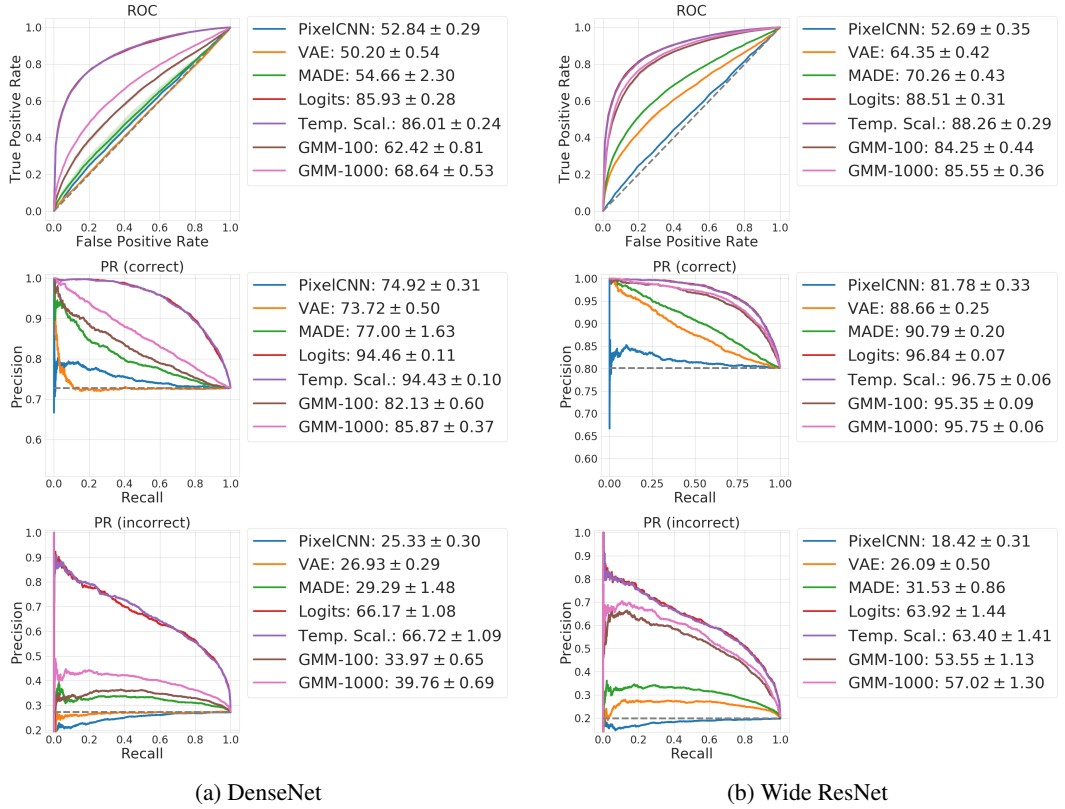

(a) DenseNet  (b) Wide ResNet

Figure 3: Comparison of ROC and PR curves for detecting classification mistakes using different generative models. Using a PixelCNN to model the image space fails at reliably detecting classification mistakes. GMMs trained on the feature space achieve better detection than other more flexible models like VAE and MAF and are comparable to temperature scaling on the WRN model.

models trained either on images or on the feature space. We find that using a GMM has similar performance than calibrated scores on the Wide ResNet but not on the DenseNet. This is explained by the fact that our DenseNet model has much lower accuracy than Wide ResNet ($72.76\%$ v. $80.22\%$) and therefore does not produce overly confident predictions. Additional results are available in the appendix.

In the next experiments, we show that although using predicted logits provides reliable detection of test set mistakes, this metric doesn't generalize to adversarial or out-of-detection samples. On the other hand, our approach to train a generative model on the feature space can be applied to these other sources of classification errors.

**Adversarial samples**   We craft adversarial samples from test samples using the Fast-Gradient Sign Method (FGSM) proposed by Goodfellow et al. (2014) and the Basic Iteration Method (BIM) (Kurakin et al., 2016). Both methods move the input image in the direction of the gradient of the loss while restraining the adversarial sample to be in a $\ell_1$ ball of ray $\epsilon_{attack}$ around the original input. This ensures that the generated adversarial sample is visually indistinguishable from the original.

Figure 4 shows that the GMM is sensitive to features extracted from adversarial samples, as they are assigned higher BPDs than clean samples. We also plot the ROC curves and corresponding AUC metrics that are obtained by using the predicted BPD to detect adversarial samples.

We compare our approach with other possible detection metrics. In particular, the method proposed by Zheng & Hong (2018) and the Mahalanobis score from Lee et al. (2018) also leverage the feature space to detect adversarial inputs. These approaches use one different model per class and therefore require labels to train while we only train one GMM in an unsupervised manner. ROC curves are shown in Figure 5 and a full comparison table with higher $\epsilon_{attack}$ values for both attacks is shown in

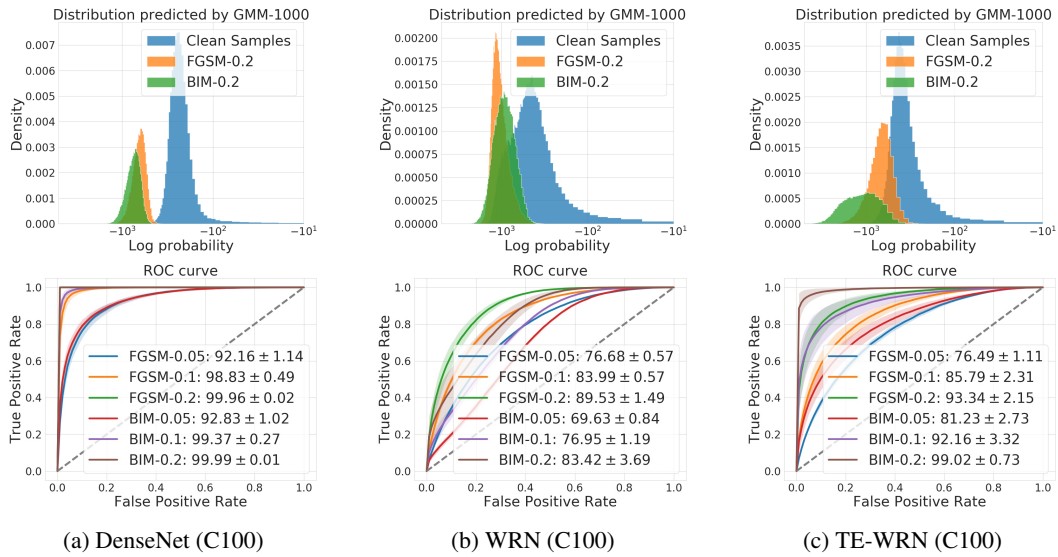

Figure 4: Top: Distribution of log-likelihood predicted on clean (blue) and adversarial samples (green and orange) by a GMM-1000. The log-likelihood of features extracted from adversarial samples is lower. The histograms are separated, which means it is possible to detect adversarial samples using the log-likelihood of their features. Bottom: ROC curve for detecting adversarial samples using predicted log-likelihood. Our method achieves a good trade-off between true positive and false positive rate, significantly improves over chance and achieves between 76% and 100% AUC depending on attack methods and models.

the appendix. Our method, using a GMM-1000 provides better detection performance of adversarial samples than calibrated and non-calibrated logit scores. Most notably, in a semi-supervised setting (Figure 5c), our method surpasses all others on attacks with low $\epsilon_{attack}$ values.

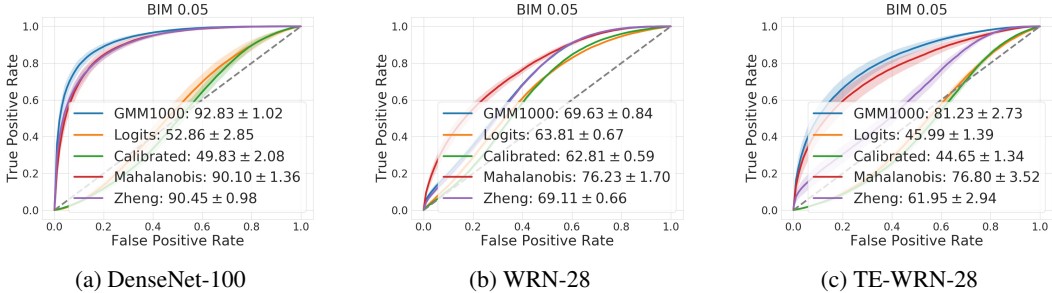

Figure 5: Comparison of ROC curves for adversarial sample detection using different metrics. Logit-based scores (Logits and Calibrated) can not reliably detect adversarial samples properly while methods that model the probability space of the feature space can (GMM1000, Mahalanobis, Zheng). Our method achieves comparable detection results as the Mahalanobis and the Zheng metric and surpasses both of them in a semi-supervised setting (Figure 5c).

**Out-of-Distribution Detection** We also test the use of feature log-likelihood values on the task of detection out-of-distribution samples. As Out-of-Distribution samples we use Random Gaussian Noise, SVHN (Netzer et al., 2011), Tiny ImageNet (Russakovsky et al., 2015), and Fashion MNIST (Xiao et al., 2017). OOD detection results are reported in Table 2 for each model we trained. Our experiments show that it is not possible to rely on calibrated probability scores for OOD detection, and that our method yields better detection results than using the Mahalanobis score in some cases. We also highlight that a PixelCNN trained on CIFAR has very poor detection results on image datasets that visually look very different from its original training set (FashionMNIST and SVHN).

This is a result of the generative model assigning higher likelihood to these OOD samples. Table 3 in the annex also shows that only calibrated scores fail to detect random gaussian noise as an OOD sample.

Table 1: OOD Detection results. Calibrated scores do not detect OOD samples well, especially samples from SVHN. On the other hand, our detection method using a GMM on the feature space is able to reliably detect out-of-distribution samples and performs better than the class-dependent Mahalanobis score even on fully-supervised models. PixelCNN assigns higher likelihoods to OOD samples that diverge clearly from the original training set and fails to detect them.

(a) Out-distribution: SVHN

| MODEL | DETECTION METHOD | AU-ROC | AU-PR (in) | AU-PR (out) |
|---|---|---|---|---|
| DenseNet-100 | GMM-1000 | $\mathbf{95.70 \pm 0.36}$ | $\mathbf{83.60 \pm 1.04}$ | $\mathbf{99.23 \pm 0.07}$ |
| | Mahalanobis | $94.38 \pm 0.39$ | $80.06 \pm 0.87$ | $98.83 \pm 0.09$ |
| | Calibrated Scores | $80.58 \pm 2.00$ | $56.60 \pm 4.74$ | $95.70 \pm 0.44$ |
| WRN-28 | GMM-1000 | $\mathbf{81.40 \pm 3.74}$ | $\mathbf{53.64 \pm 6.02}$ | $\mathbf{96.45 \pm 0.87}$ |
| | Mahalanobis | $75.86 \pm 2.72$ | $\mathbf{47.40 \pm 6.42}$ | $93.96 \pm 0.61$ |
| | Calibrated Scores | $78.64 \pm 1.70$ | $\mathbf{56.23 \pm 3.00}$ | $95.05 \pm 0.52$ |
| TE-WRN-28 | GMM-1000 | $\mathbf{72.45 \pm 7.41}$ | $\mathbf{40.62 \pm 7.64}$ | $\mathbf{93.47 \pm 2.03}$ |
| | Mahalanobis | $49.46 \pm 5.52$ | $17.32 \pm 5.19$ | $86.54 \pm 1.36$ |
| | Calibrated Scores | $62.10 \pm 3.97$ | $\mathbf{35.46 \pm 4.82}$ | $89.82 \pm 1.10$ |
| Model-Agnostic | PixelCNN | $20.18 \pm 0.41$ | $7.33 \pm 0.10$ | $75.97 \pm 0.10$ |

(b) Out-distribution: Tiny ImageNet

| MODEL | DETECTION METHOD | AU-ROC | AU-PR (in) | AU-PR (out) |
|---|---|---|---|---|
| DenseNet-100 | GMM-1000 | $\mathbf{96.75 \pm 0.45}$ | $\mathbf{96.78 \pm 0.42}$ | $\mathbf{96.67 \pm 0.51}$ |
| | Mahalanobis | $95.26 \pm 0.61$ | $95.68 \pm 0.55$ | $94.57 \pm 0.70$ |
| | Calibrated Scores | $74.00 \pm 2.01$ | $73.49 \pm 4.37$ | $71.45 \pm 1.09$ |
| WRN-28 | GMM-1000 | $\mathbf{84.98 \pm 1.86}$ | $\mathbf{86.32 \pm 1.95}$ | $\mathbf{82.45 \pm 1.92}$ |
| | Mahalanobis | $80.35 \pm 2.92$ | $82.72 \pm 2.62$ | $76.01 \pm 3.91$ |
| | Calibrated Scores | $82.64 \pm 1.65$ | $\mathbf{87.46 \pm 1.43}$ | $79.40 \pm 1.56$ |
| TE-WRN/C100 | GMM-1000 | $67.94 \pm 2.11$ | $67.82 \pm 1.44$ | $68.59 \pm 2.84$ |
| | Mahalanobis | $\mathbf{70.63 \pm 2.46}$ | $\mathbf{70.66 \pm 1.83}$ | $\mathbf{70.08 \pm 3.10}$ |
| | Calibrated Scores | $56.41 \pm 0.96$ | $59.22 \pm 1.43$ | $54.01 \pm 0.49$ |
| Model-Agnostic | PixelCNN | $82.05 \pm 0.14$ | $78.63 \pm 0.16$ | $81.87 \pm 0.19$ |

(c) Out-distribution: Fashion-MNIST

| MODEL | DETECTION METHOD | AU-ROC | AU-PR (in) | AU-PR (out) |
|---|---|---|---|---|
| DenseNet-100 | GMM-1000 | $\mathbf{93.23 \pm 1.31}$ | $\mathbf{89.29 \pm 2.75}$ | $\mathbf{98.30 \pm 0.36}$ |
| | Mahalanobis | $\mathbf{94.48 \pm 0.41}$ | $\mathbf{88.95 \pm 1.04}$ | $\mathbf{98.12 \pm 0.15}$ |
| | Calibrated Scores | $85.92 \pm 1.72$ | $65.72 \pm 3.05$ | $96.60 \pm 0.5$ |
| WRN-28 | GMM-1000 | $\mathbf{87.32 \pm 2.12}$ | $\mathbf{69.72 \pm 3.45}$ | $\mathbf{96.82 \pm 0.67}$ |
| | Mahalanobis | $79.33 \pm 1.90$ | $57.26 \pm 2.33$ | $93.7 \pm 0.80$ |
| | Calibrated Scores | $\mathbf{86.01 \pm 0.99}$ | $\mathbf{70.37 \pm 1.49}$ | $\mathbf{96.24 \pm 0.39}$ |
| TE-WRN-28 | GMM-1000 | $66.95 \pm 3.85$ | $36.79 \pm 4.06$ | $\mathbf{91.29 \pm 1.34}$ |
| | Mahalanobis | $63.96 \pm 1.73$ | $32.71 \pm 2.00$ | $89.88 \pm 0.75$ |
| | Calibrated Scores | $\mathbf{70.11 \pm 2.27}$ | $\mathbf{43.56 \pm 3.38}$ | $\mathbf{91.72 \pm 0.87}$ |
| Model-Agnostic | PixelCNN | $0.71 \pm 0.09$ | $7.51 \pm 0.00$ | $67.70 \pm 0.02$ |

## 5 CONCLUSION

Using statistical hypothesis testing we provided a general characterization of inputs that lead to classification mistakes by deep neural networks. With a simple Gaussian Mixture Model, we model the distribution of the feature space learned by a classifier and verified that features extracted from inputs consistently lie outside of the training distribution and can be detected by their low predicted log probability. Compared to other score-based methods, our characterization holds for a variety of classification failure modes in deep neural networks: adversarial sample detection, out-of-distribution detection and test time classification mistakes.

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

# A    ADDITIONAL DETECTION RESULTS

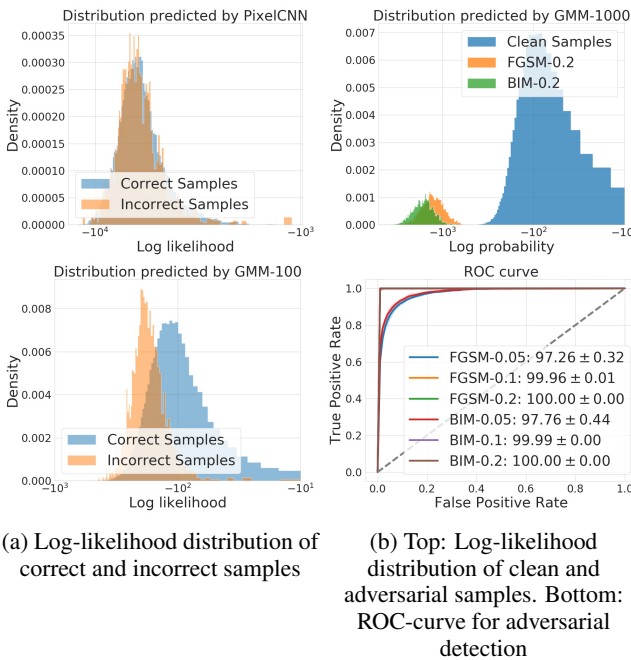

(a) Log-likelihood distribution of correct and incorrect samples

(b) Top: Log-likelihood distribution of clean and adversarial samples. Bottom: ROC-curve for adversarial detection

Figure 6: Additional results for model trained on CIFAR-10. Left: comparison of log-likelihood distributions. PixelCNN is not able to distinguish correct samples from incorrect ones while a GMM trained on the feature space can. Middle: a GMM trained on the feature space can detect adversarial samples reliably.

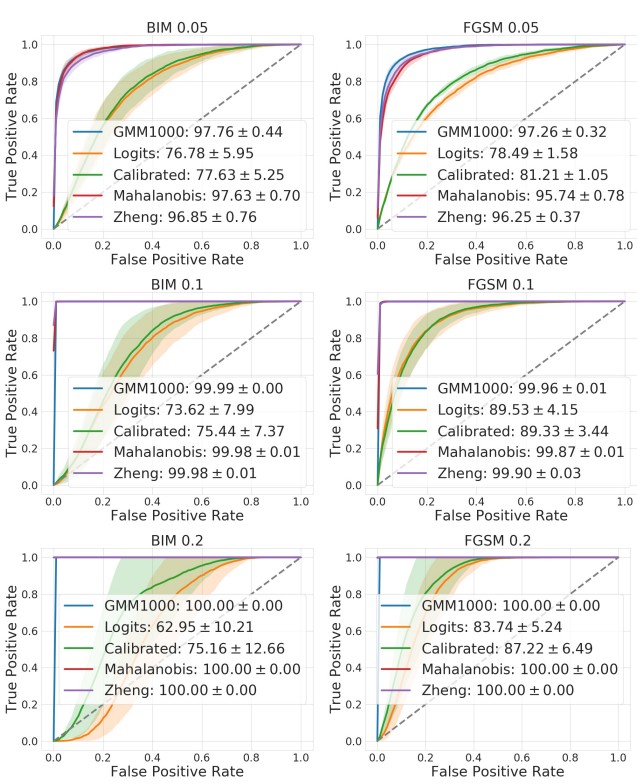

Figure 7: Comparison of ROC curves for FGSM and BIM adversarial sample detection for CIFAR-10 model.

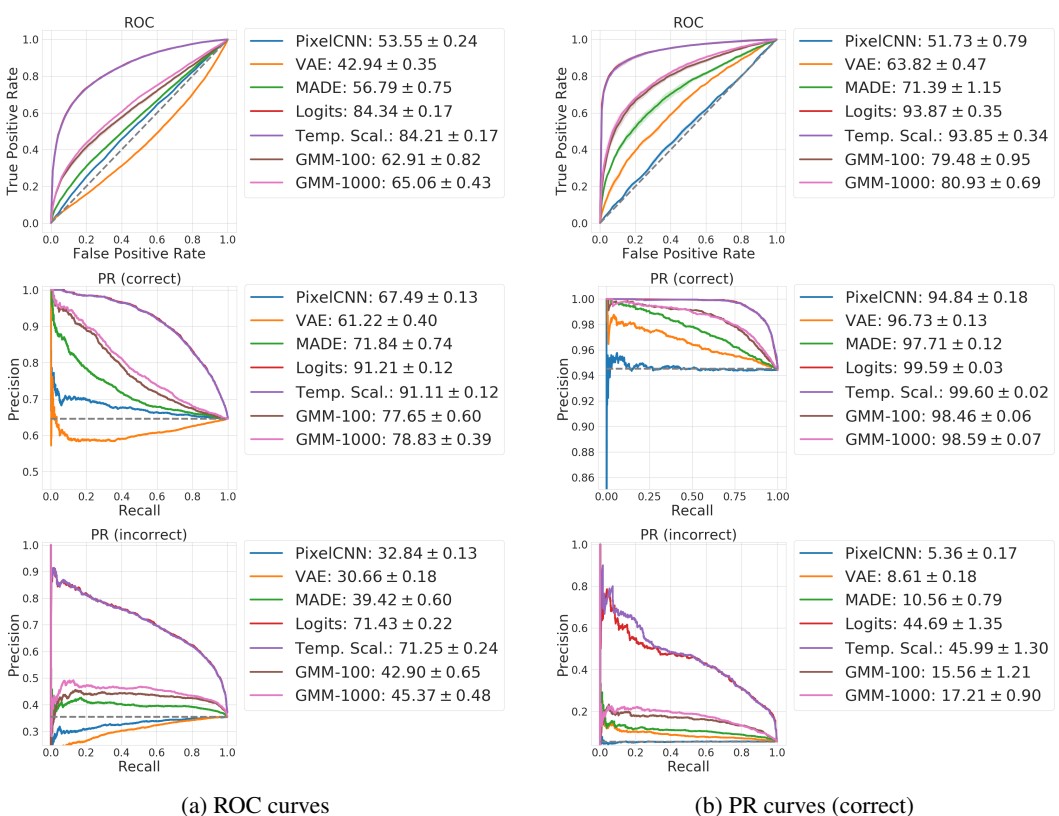

(a) ROC curves           (b) PR curves (correct)

Figure 8: Additional comparison of ROC and PR curves for detecting classification mistakes using different generative models. Using a PixelCNN to model the image space fails at reliably detecting classification mistakes. GMMs trained on the feature space achieve better detection than other more flexible models like VAE and MAF and are comparable to temperature scaling on the WRN model.

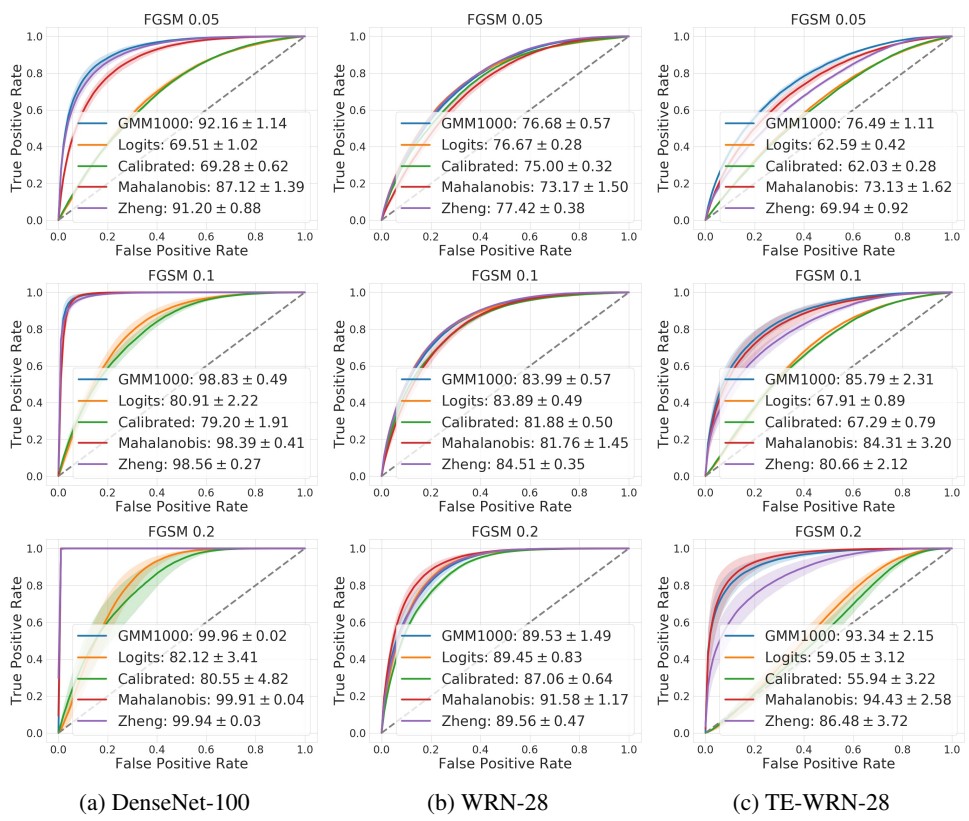

Figure 9: Additional comparison of ROC curves for FGSM adversarial sample detection for CIFAR-100 models

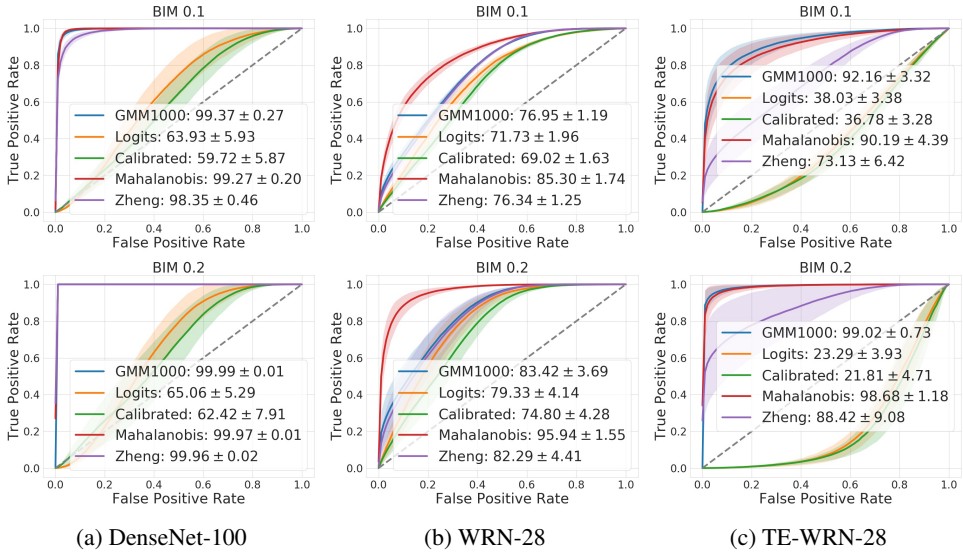

Figure 10: Additional comparison of ROC curves for BIM adversarial sample detection for CIFAR-100 models.

Table 2: OOD Detection results for CIFAR-10 model. Calibrated scores do not detect OOD samples well, especially gaussian noise samples. PixelCNN assigns higher likelihoods to OOD samples that diverge clearly from the original training set and fails to detect them.

(a) Out-distribution: SVHN

| MODEL | DETECTION METHOD | AU-ROC | AU-PR (in) | AU-PR (out) |
|---|---|---|---|---|
| DenseNet-100 | GMM-1000 | **99.35 ± 0.21** | **96.46 ± 0.90** | **99.91 ± 0.03** |
| | Mahalanobis | 98.82 ± 0.44 | 93.06 ± 2.37 | 99.83 ± 0.06 |
| | Calibrated Scores | 92.00 ± 3.02 | 82.48 ± 4.90 | 98.29 ± 0.70 |
| Model-Agnostic | PixelCNN | 19.73 ± 0.72 | 7.29 ± 0.08 | 76.06 ± 0.21 |

(b) Out-distribution: Tiny ImageNet

| MODEL | DETECTION METHOD | AU-ROC | AU-PR (in) | AU-PR (out) |
|---|---|---|---|---|
| DenseNet-100 | GMM-1000 | **98.88 ± 0.28** | **98.90 ± 0.25** | **98.89 ± 0.29** |
| | Mahalanobis | 98.03 ± 0.39 | 98.07 ± 0.37 | 98.03 ± 0.42 |
| | Calibrated Scores | 94.66 ± 0.50 | 95.66 ± 0.36 | 93.49 ± 0.78 |
| Model-Agnostic | PixelCNN | 85.01 ± 0.16 | 80.11 ± 0.25 | 86.22 ± 0.19 |

(c) Out-distribution: Gaussian Noise

| MODEL | DETECTION METHOD | AU-ROC | AU-PR (in) | AU-PR (out) |
|---|---|---|---|---|
| DenseNet-100 | GMM-1000 | **100** | **100** | **100** |
| | Mahalanobis | **100** | **100** | **100** |
| | Calibrated Scores | 77.85 ± 18.34 | 86.57 ± 11.61 | 68.28 ± 16.81 |
| Model-Agnostic | PixelCNN | 100 | 100 | 100 |

(d) Out-distribution: Fashion-MNIST

| MODEL | DETECTION METHOD | AU-ROC | AU-PR (in) | AU-PR (out) |
|---|---|---|---|---|
| DenseNet-100 | GMM-1000 | **97.47 ± 1.10** | **91.28 ± 3.91** | **99.52 ± 0.20** |
| | Mahalanobis | **98.38 ± 0.87** | **95.07 ± 2.71** | **99.67 ± 0.18** |
| | Calibrated Scores | 93.71 ± 1.83 | 85.72 ± 3.97 | 98.53 ± 0.48 |
| Model-Agnostic | PixelCNN | 0.61 ± 0.16 | 7.51 ± 0.00 | 67.68 ± 0.03 |

Table 3: Out-distribution detection results for CIFAR-100 models on Gaussian Noise. Calibrated scores is the only method failing at detecting gaussian noise inputs.

| MODEL | DETECTION METHOD | AU-ROC | AU-PR (in) | AU-PR (out) |
|---|---|---|---|---|
| DenseNet-100 | GMM-1000 | **100** | **100** | **100** |
| | Mahalanobis | **100** | **100** | **100** |
| | Calibrated Scores | 70.04 ± 20.28 | 80.36 ± 13.99 | 61.56 ± 15.92 |
| WRN-28 | GMM-1000 | **99.96 ± 0.03** | **99.96 ± 0.03** | **99.94 ± 0.05** |
| | Mahalanobis | **99.99 ± 0.01** | **100** | **99.98 ± 0.02** |
| | Calibrated Scores | 88.27 ± 6.07 | 92.61 ± 3.67 | 78.46 ± 9.80 |
| TE-WRN-28 | GMM-1000 | **100** | **100** | **100** |
| | Mahalanobis | **100** | **100** | **100** |
| | Calibrated Scores | 38.28 ± 16.31 | 59.02 ± 11.95 | 41.07 ± 5.55 |
| Model-Agnostic | PixelCNN | 100 | 100 | 100 |

# B PURIFICATION

## B.1 METHOD

The purification process aims at moving the feature $\mathbf{F}$ extracted by the classifier to a low BPD region. This can be formulated as a joint optimization problem where we want to find features $\mathbf{F}$ with minimal BPD, while being close to the initial extracted features $\mathbf{F}_{ref}$.

$$\mathbf{F}^{\star} = \arg \min_{\mathbf{F}} BPD(\mathbf{F}) + \nu \|\mathbf{F} - \mathbf{F}_{ref}\|_2^2 \tag{8}$$

$\nu$ is a hyperparameter that defines how close the new feature should be to the initial one. As the objective is not convex and there is no close form solution for stationary points, we use gradient descent with regards to $\mathbf{F}$ to minimize the objective function.

$$\mathbf{F} := \mathbf{F} - \epsilon(\nabla_{\mathbf{F}} BPD(\mathbf{F}) + 2\nu(\mathbf{F} - \mathbf{F}_{ref})) \tag{9}$$

## B.2 PURIFICATION RESULTS

Purification of features is performed with 100 iterations of gradient descent steps to optimize the objective function. We test the performance of purification for both classification and semi-supervised classification tasks on CIFAR-100.

We report the accuracy on validation and test set obtained after purification with different GMMs and for different values of learning rates $\epsilon$ and regularization strength $\nu$ in Table 4. For classification, our networks are DenseNet (DN-100) and Wide ResNet (WRN-28). For semi-supervised classification, we apply temporal ensembling to wide ResNet (TE-WRN-28). Our results show that this purification procedure is able to correct classification mistakes on previously unseen samples and results in an accuracy gain for the model without the need to retrain. However the purification method also leads to new classification mistakes, which means that the net improvement on the accuracy reaches $0.6\%$ on the DenseNet model at most.

Table 4: Validation and test classification accuracy obtained after purification for DenseNet-100, Wide Resnet-28 and TE-Wide Resnet-28. Purification increases the test accuracy by up to $0.6\%$ on the DenseNet model.

| GMM | $\epsilon$ | $\nu$ | DN-100 | | WRN-28 | | TE-WRN-28 | |
|---|---|---|---|---|---|---|---|---|
| | | | Val | Test | Val | Test | Val | Test |
| Original | - | - | 73.12 | 72.74 | 80.34 | 80.10 | 65.13 | 64.51 |
| 1000 | 0.1 | 0 | 73.78 | 73.39 | 80.01 | 79.73 | 65.21 | 64.61 |
| | | 0.01 | **73.78** | **73.39** | 80.03 | 79.75 | 65.24 | 64.60 |
| | | 0.1 | 73.74 | 73.28 | 80.28 | 79.87 | **65.27** | **64.60** |
| | | 1.0 | 73.24 | 72.89 | 80.40 | 80.11 | 65.16 | 64.51 |
| | 0.01 | 0 | 73.33 | 72.95 | 80.40 | 80.10 | 65.18 | 64.51 |
| | | 0.01 | 73.33 | 72.95 | 80.40 | 80.10 | 65.18 | 64.51 |
| | | 0.1 | 73.30 | 72.95 | 80.38 | 80.10 | 65.17 | 64.51 |
| | | 1.0 | 73.22 | 72.86 | **80.40** | **80.11** | 65.16 | 64.51 |

# C EXPERIMENTAL SETUP

**Dataset and preprocessing** We trained on CIFAR-10 and CIFAR-100 Krizhevsky (2009) with 5,000 images held-out validation images. Inputs were preprocessed with per-channel standardization before training.

**DenseNet** We use bottleneck layers and compression rate $\theta = 0.5$, growth rate $k = 12$ and depth $L = 100$. The model is trained with batch size 64 for 300 epochs with a learning rate 0.1, dropout rate 0.2 and $L_2$ regularization weight $1e^{-4}$. We use ReLU non-linearities except for the last layer where we use a $tanh$ non-linearity to ensure the extracted features are bounded. For optimization, we use Stochastic Gradient Descent with a Nestrov momentum of 0.9. The learning rate is divided by 10 at epoch 150 and 175.

**Wide ResNet**    Wide ResNet Zagoruyko & Komodakis (2016) is trained with growth rate $k = 10$ and depth $L = 28$ and batch size $100$ for $200$ epochs, with a learning rate $0.1$, dropout rate $0.3$ and $L_2$ regularization weight $5e^{-4}$. Data augmentation is applied during training with random translation by up to 2 pixels and random horizontal flips.

**Temporal Ensembling**    For the semi-supervised setting, we only keep 100 samples per label in the train set. We train a Wide ResNet using Temporal Ensembling with a maximum weight decay of $100$.

**PixelCNN**    The PixelCNN model is trained with the PixelCNN++ ameliorations from Salimans et al. (2017) for our experiments. The model is trained for $5000$ epochs with dropout rate $0.5$ and learning rate $1e^{-4}$.

**VAE**    The VAE is trainer for $1000$ epochs with a learning rate of $0.001$ and decay rate of $0.9995$. The encoder and decoder architecture are fully connected layers with ReLU non-linearities, one hidden layer of size $512$ and latent dimension of $128$. The model was trained with Adam.

**MAF**    The Masked Autoregressive Flow model is trained for $1000$ epochs with a learning rate of $0.01$ and batch size $32$ using Adam Optimizer. We used a 5-layer MADE model with hidden layer size of $128$.

**Temperature Scaling**    The temperature for Temperature Scaling is optimized using the L-BFGS-B optimization algorithm with a maximum of 100 iterations. We use ECE with $B = 10$ bins to evaluate the success of the calibration.

