# OpenReview forum: "Understanding Classifiers with Generative Models"
_ICLR.cc/2021/Conference — Reject_

### Official Review · AnonReviewer3 · 2020-10-21
**Interesting observation but not enough**

**Rating:** 5
**Confidence:** 4

**Review:**

The main novel contribution is that the authors use generative models for several tasks but on the feature space and not the input pixel space. They show interesting behavior that mistakes tend to have low density in feature space, which isn't obvious as the NN that computes the features can map them wrongly to a high likelihood location.
However, the authors claims of usefulness is not properly demonstrated in the experiences.

- The authors claim several times that they perform "rigorous hypothesis testing" (including the abstract and conclusions so not a minor point in the paper) but it doesn't really mean that the mistakes can be identified as one can understand from the text. It just means that the distributions are distinct (but can have large overlap as can be seen in fig.2)
- While GMM on feature space are better then other models on feature space and pixel space, it isn't convincing that the error detection can be of any use.
- Adversarial detection was only tested on adversaries that do not try to fool the detector. As was shown in "Adversarial Examples Are Not Easily Detected: Bypassing Ten Detection Methods" by Carlini&Wagner this can lead to wrong detection claims.

---

> ### Author Response · Authors · 2020-11-24
> **Rebuttal**
>
> Thank you for taking the time to review our paper.
> We'd like to address a particular point of your review:
>
> Remark: The authors claim several times that they perform "rigorous hypothesis testing" (including the abstract and conclusions so not a minor point in the paper) but it doesn't really mean that the mistakes can be identified as one can understand from the text. It just means that the distributions are distinct (but can have large overlap as can be seen in fig.2)
>
> Answer: Rigorous Hypothesis Testing refers to the statistical testing we describe in the paper. We do not claim that  mistakes can be identified from the hypothesis testing but merely that the statistical testing shows the distributions are distinct. This is not a trivial observation as we show in the paper that the distributions learned by a PixelCNN trained on the input space are not distinct. The statistical test gives us a measure of how distinct the distributions are.

---

### Official Review · AnonReviewer2 · 2020-10-27

**Rating:** 4
**Confidence:** 4

**Review:**

**Update after rebuttal:** The author rebuttal clarified some minor issues for me, but it did nothing to address my main concern, which is that very similar methods have been proposed before. I'm therefore keeping my score the same.

---------------------------------------------
This paper proposes a simple method for out-of-distribution detection. The basic idea is to fit a GMM to training examples in the feature space (as opposed to the pixel space). The experiments are generally rigorous and well executed, however my main problem with the paper is that it seems a bit too incremental to justify another paper. As the authors correctly point out very similar methods have already been proposed before (Zheng & Hong, 2018; Lee et al., 2018).

On page 2, the authors claim “Zheng & Hong (2018) and Lee et al. (2018) train a conditional generative model on the feature space learned by the classifier and derive a confidence score based on the Mahalanobis distance between a test sample and its predicted class representation”. However, this is not correct, Zheng and Hong (2018) don’t use the Mahalanobis metric. They use the exact same likelihood-based criterion in this paper, except they do this on a class-conditional basis as the authors correctly point out, but that seems like a small difference (as labels will obviously be available for the training data in a supervised setting, and the method can be easily adapted to the semi-supervised setting with some tweaks). I just don't see how this small difference could have a huge effect on performance.

The authors do have some comparisons with these earlier methods, but I’m wondering if these comparisons are done fairly: for example Zheng and Hong (2018) have a similar K parameter in their GMM model, what is the value used for that parameter in this paper? Is it the same as the one used in the class-agnostic GMMs? Have you tried tuning that parameter for these earlier models? How was the threshold parameter chosen? As far as I can see, these important experimental details are not discussed at all anywhere in the paper. Similarly, the semi-supervised setting in Figure 5c is not explained at all. How exactly does it work for the different models shown in that figure? Also, why does the Mahalanobis metric seem to work much better than the GMM in Figure 5b?

Another question I have is whether the authors have tried using features other than the final embedding layer features (and possibly a combination of features from multiple layers: something like this was done before in the deep k-nn paper by Papernot & McDaniel: https://arxiv.org/abs/1803.04765).

More minor comments:

Page 6: please make sure you mention what BPD means (binned probability distribution), I don’t think this is as commonly known as CDF or PDF.

Typos: “Aims at modeling confidence score that are” (p. 2), “task of detection out of distribution samples” (p. 7).

---

> ### Author Response · Authors · 2020-11-24
> **Rebuttal**
>
> Thank you for your review. Below are questions we would like to address:
>
> * Question: The authors do have some comparisons with these earlier methods, but I’m wondering if these comparisons are done fairly: for example Zheng and Hong (2018) have a similar K parameter in their GMM model, what is the value used for that parameter in this paper? Is it the same as the one used in the class-agnostic GMMs?
>
> * Answer: To make the comparison fair between Zheng and our method, we used a **GMM-1000 for our method and a GMM-10 per class ** for Zheng’s method so that we have the same number of parameters for each method.
>
> * Question: How was the threshold parameter chosen? As far as I can see, these important experimental details are not discussed at all anywhere in the paper.
>
> * Answer: We do not choose the threshold parameter explicitly in the experiments, instead reporting the ROC and PR curves.
>
> * Question: Similarly, the semi-supervised setting in Figure 5c is not explained at all. How exactly does it work for the different models shown in that figure?
>
> * Answer: The Temporal Ensembling setting is explained in the first paragraph of section 4 on page 5. We explain how many samples per labels we keep and cite the paper from which the Temporal Ensembling is implemented.
>
> * Page 6: please make sure you mention what BPD means (binned probability distribution), I don’t think this is as commonly known as CDF or PDF.
>
> * Answer: Apologies for the confusion. BPD actually stands for Bits per Dimension and is a scaled version of log-likelihood that takes into account the size of features. This will be corrected in the final version. (BPD will not be mentioned and we will use log-likelihood instead)

---

### Official Review · AnonReviewer4 · 2020-10-28
**Weak recommendation to accept**

**Rating:** 6
**Confidence:** 4

**Review:**

The authors provide a new method for detecting when deep networks are likely to fail and demonstrate through extensive experimentation its accuracy against generalization errors, out of distribution samples and adversarial attacks. The method builds on prior Mahalanobis metric of (Kimin Lee, et al., A unified framework for detecting out-of-distribution and adversarial samples. 2018) in two respects. First the authors use a single GMM fit to the model parameters that is class agnostic rather than a set of GMMs for each class, thus making it suitable for application in semi-supervised datasets. Second, the authors show ability to detect instances from a test set that are likely to cause a misclassification due to a failure to generalize. Surprisingly, the proposed approach performs better in most cases than the prior Mahalanobis approach even though it requires less information (no labels).

Pros:

1. Well written and organized paper.
2. More general and simpler approach than prior art
3. Extensive empirical results that are competitive with or improved over prior art.

Cons:

1. This paper is very similar to and provides only a relatively small incremental improvement over prior art (Lee, et al.)
2. Like the Mahalanobis method, the proposed hypothesis testing method requires fitting a GMM to an "incorrect" distribution. Please make it clear what this distribution is for the experiments. It seems this would require knowledge of the type of attack, or the out-distribution making it non-blind and unrealistic for real-world out-of-distribution or adversarial attack scenarios.

---

### Official Review · AnonReviewer1 · 2020-10-29
**Interesting study of learned neural features, with premature conclusion due to experiments on limited data**

**Rating:** 5
**Confidence:** 4

**Review:**

Motivated by the need to assess a classifier's confidence on its decision,
the paper proposes to use predicted likelihood of the learned features
of an unknown sample, and shows that samples with low feature
likelihood tend to receive a wrong decision.
Furthermore, the paper argues that using a GMM do model the feature
distributions is better than other methods like VAE or AR flow.

This study is interesting in the sence that it attempts to analyze the
distributions of the learned features, which is much needed for better
understanding of what a deep neural net attempts to do.
The findings appear to make good sense as the network training process
is designed to push samples of like-class towards tight clusters when
mapped to the learned feature space.  Those lying in the outskirts signal
difficulties in this process,  therefore they are likely to get
erroneous decisions.   The feature distributions are modeled
class-blind, so that the prediction can be applied to unseen cases
without class labels.

The work can be improved by performing this analysis at different stages
of a deep network,  as one expects that the levels closer to the
output layer demonstrate more of such clustering effect.  It
will be useful to confirm with this analysis.

Other than showing numerical results, it will be more convincing if
example images are shown that are identified by this method to
have unreliable decisions by the classifier, and what error the classifier
makes on them.

A more significant weakness is that the experiments are done only with
an image classification problem, and a single dataset.  This makes the
conclusion somewhat premature,  as images of physical objects tend to
form good patterns.  Is the GMM estimation good just because the
data happen to be well clustered?  Will this conclusion be confirmed
by classification tasks on other types of data, e.g. text?
What if the class labels are scrambled?
What makes some network settings better than others in learning
such confidence-suggestive features?

Misc.:

p.1, line 4 from bottom,
"... capture the distribution of the learned feature space" ->
"capture the class-conditional distributions in the learned feature space"

p.4, line 5,
"... state-of-the-art models assigning higher likelihoods to samples
... " ->
What kind of models does this refer to?  What is the model supposed to
do?  Why do they assign higher likelihoods to out-of-distribution samples?
There is a lot to be filled in here; citing an external reference is
not enough.

p.6, line 7 from bottom,
"... assigned higher BPDs ...", please expand the acronym BPD at its
first mention.

p.8, conclusion,
"... verified that features extracted from inputs consistently lie
outside of the training distribution and can be detected by their low
predicted log-probability."   This sentence is garbled.  What have
you verified?  Do you mean to say this for a special type of input?
What consistently lie outside of what?

---

> ### Author Response · Authors · 2020-11-24
> **Rebuttal**
>
> Thank you for your feedback. Below are a couple of specific points of the review we would like to address.
>
> * Remark: p.1, line 4 from bottom, "... capture the distribution of the learned feature space" -> "capture the class-conditional distributions in the learned feature space"
>
> * Answer: Please note the the distribution we learn with the GMM are not class-conditional since we’re not using class labels to train the GMM on the feature space.
>
>
> * Remark: p.4, line 5, "... state-of-the-art models assigning higher likelihoods to samples ... " -> What kind of models does this refer to? What is the model supposed to do? Why do they assign higher likelihoods to out-of-distribution samples? There is a lot to be filled in here; citing an external reference is not enough.
>
> * Answer: We refer here to generative models that learn the probability distribution of their training dataset. (For example a PixelCNN trained on CIFAR10). Given an input sample, the model yields an estimate of the sample’s log-likelihood under the data distribution. Out-of-distribution samples that differ from the training data should be predicted to have lower log-likelihood.
>
> * Remark: Question: p.6, line 7 from bottom, "... assigned higher BPDs ...", please expand the acronym BPD at its first mention.
>
> * Answer: Apologies for the confusion. BPD refers to “Bit-per-dimension”, a scaled measure of log-likelihood that takes into account the size of the feature. This should be replaced by “assigned lower log-likelihood”.
>
> * Remark: p.8, conclusion, "... verified that features extracted from inputs consistently lie outside of the training distribution and can be detected by their low predicted log-probability." This sentence is garbled. What have you verified? Do you mean to say this for a special type of input? What consistently lie outside of what?
>
> * Answer: The sentence should be corrected to “verified that features extracted from inputs **leading to classification mistakes**, consistently lie outside of the training distribution and can be detected by their low predicted log-probability.”

---

### Decision · Program_Chairs · 2021-01-07
**Final Decision**

**Decision:**

Reject

**Comment:**

As several reviewers pointed out, the contribution is  too incremental from previous work.